# Modelling Fibre-Reinforced Concrete for Predicting Optimal Mechanical Properties

**DOI:** 10.3390/ma16103700

**Published:** 2023-05-12

**Authors:** Hamad Hasan Zedan Khalel, Muhammad Khan

**Affiliations:** School of Aerospace, Transport and Manufacturing, Cranfield University, Building 50, College Road, Cranfield MK43 0AL, UK; h.khalel@cranfield.ac.uk

**Keywords:** steel fibre, plastic fibre, reinforced concrete, mechanical properties

## Abstract

Fibre-reinforced cementitious composites are highly effective for construction due to their enhanced mechanical properties. The selection of fibre material for this reinforcement is always challenging as it is mainly dominated by the properties required at the construction site. Materials like steel and plastic fibres have been rigorously used for their good mechanical properties. Academic researchers have comprehensively discussed the impact and challenges of fibre reinforcement to obtain optimal properties of resultant concrete. However, most of this research concludes its analysis without considering the collective influence of key fibre parameters such as its shape, type, length, and percentage. There is still a need for a model that can consider these key parameters as input, provide the properties of reinforced concrete as output, and facilitate the user to analyse the optimal fibre addition per the construction requirement. Thus, the current work proposes a Khan Khalel model that can predict the desirable compressive and flexural strengths for any given values of key fibre parameters. The accuracy of the numerical model in this study, the flexural strength of SFRC, had the lowest and most significant errors, and the MSE was between 0.121% and 0.926%. Statistical tools are used to develop and validate the model with numerical results. The proposed model is easy to use but predicts compressive and flexural strengths with errors under 6% and 15%, respectively. This error primarily represents the assumption made for the input of fibre material during model development. It is based on the material’s elastic modulus and hence neglects the plastic behaviour of the fibre. A possible modification in the model for considering the plastic behaviour of the fibre will be considered as future work.

## 1. Introduction

The construction industry uses a wide range of composite materials, the most common of which is concrete. Concrete offers good strength and durability in constructing structures, but is brittle in nature [1,2]. This is better for structures working under compressive loads. However, in applications where structures are under bending or tension, it is deemed necessary to reinforce concrete with materials that can provide the required ductility and not reduce the most needed compressive strength [3,4,5]. Due to this reason, academic and industrial domain researchers have used fibres of small sizes with good flexural and tensile properties as a constituent of concrete [4,6,7,8,9,10,11]. In the past, steel fibres (SF) were added to recycled aggregate concrete (RAC) and demonstrated an increase in tensile strength, elastic modulus, and post-cracking behaviour [12,13]. The researchers found that SFRC suits the structures that experience loads over the serviceability limit state in shear, bending, and impact or dynamic forces under seismic or cyclic activity [14,15]. It was found that the percentage of fibre by volume has little effect on compressive strength [15,16]. Utilizing fly ash and/or PVA fibre refines the pore structure, thereby enhancing frost resistance. In contrast, MgO and SRA are less effective than PVA fibre and fly ash at refining the pores, resulting in smaller and relatively weakened frost resistance [17]. There is no correlation between the compressive strength and abrasion resistance of hydraulic concretes containing MgO and/or PVA fibre and the pore structure parameters and pore surface fractal dimensions [18].

Granulated blast furnace slag was used as a fibre, and the obtained properties were plotted using multivariable linear regression. It was observed that the percentage of fibre by weight significantly impacts compressive strength [19,20,21,22,23]. According to a statistical study, the synergistic effect of the linear term of the R-ratio has a significant impact on early compressive strength [24]. Hamed et al. employed statistical tools to predict the thermo-mechanical properties of concrete reinforced with rubber aggregate. They used the Taylor diagram and meant absolute errors to discuss the obtained properties [25,26,27,28,29,30,31]. Other fibres were tested for the tensile strength of reinforced concrete, and multiple linear regression was used to model the test findings [32,33,34,35]. According to the published statistical analysis, fibre hybridization positively influences flexural strength, depending on the fibre type and volume fraction [36]. The ANN model and the regression model achieved a good prediction of the IST strength of SFRC in evaluation [37]. Deng et al. proposed an empirical constitutive model to describe the stress-strain relationship and damage accumulation in hybrid fibre-reinforced concrete (HFRC). The concrete was subjected to uniaxial cyclic tensile load and the model used volume fraction and aspect ratio of fibre as inputs. They also discussed plastic strain, stiffness deterioration, and damage index of the reinforced concrete with the help of their model. The model predictions agreed with the test results [38,39,40].

The fibre reinforcement was also modelled with numerical methods to determine its influence on the reinforced concrete. Lee and Fenves proposed a model of concrete damage plasticity, which is considered the fundamental contribution to analyses of the concrete properties with and without fibre reinforcement [41,42]. Later, researchers used this model in Abaqus and evaluated the concrete properties under shear loads in column construction [43]. Revanna et al. applied a CDP-based FEA model to validate a specific reinforced concrete beam experiment, concluding that the behaviour of the beam could be predicted [44]. When using the CDP model (CDPM) in Abaqus, it has been recommended to use two stress-strain curves under compressive and tensile behaviour. The suggested material model can also explain the propagation of cracks and the post-cracking behaviour of reinforced concrete structures [45,46,47,48]. Other studies have presented empirical and numerical models for predicting the flexural behaviour and compressive strength of fibre-reinforced polymer (FRP)-reinforced concrete. However, these models are highly complex [49,50,51,52,53]. MLR and CDP have recently been utilized to forecast the behaviour of reinforced concrete (RC) elements, mainly where code requirements are unavailable. MLR outperforms CDP because it can create accurate prediction models with a limited database.

However, most of the above-mentioned research concludes their analysis without considering the collective influence of key fibre parameters such as its shape, type, length, and percentage. There is still a need for a model that can consider these key parameters as input, provide the properties of reinforced concrete as output, and facilitate the user to analyze the optimal fibre addition per the construction requirements. Thus, the current work proposes a Khan Khalel model that can predict the desirable compressive and flexural strengths for any given values of key fibre parameters. Statistical tools are used to develop and validate the model with numerical results. The proposed model is easy to use but predicts compressive and flexural strengths with errors under 6% and 15%, respectively. This error primarily represents the assumption made for the input of fibre material during model development. It is based on the material’s elastic modulus and hence neglects the plastic behaviour of the fibre. A possible modification in the model for considering the plastic behaviour of the fibre will be considered as future work.

## 2. Materials and Methods

### 2.1. Materials

Portland EMC II composite cement and normal coarse and fine aggregates were used in this study. Sieve examination of the aggregates utilized demonstrated that the aggregate is suitable for creating the concrete mixture. Flowed SCC superplasticizer was added to improve the workability of the concrete so that it would have good usability and flowability. The concrete was extremely workable, with excellent cohesiveness, no segregation, and low bleed water [54]. Novocon^®^ FE-1050 steel fibres, Novocon^®^ XR-1050 steel fibres, Enduro^®^ Fiber high-performance polymer, and Enduro^®^ Mirage (which is 100% virgin copolymer fibre) were employed. The fibres were added to concrete in a range of percentages (0.5%, 1%, 1.5%, and 2%) and lengths (20 mm, 30 mm, and 40 mm). Table 1 shows the characteristics of the various fibre types, the reason for choosing these fibres to find the optimal value of the fibre by using different shapes and percentage. Figure 1 presents images of the different types of fibres used in this study.

### 2.2. Sample Preparation

A slump target in the range from 3 cm to 6 cm and a specific strength of 30 N/mm^2^ for 28 days were set (see Appendix A, Figure A1). The prepared concrete blend had a water-cement (w/c) ratio of 0.52. The amounts for mixing 1 m^3^ of control concrete are shown in Table 2. Reinforced concrete mixers were created using a one-axis horizontal mixer. This study included 49 FRC mixes, together with a control mix. For instance, type-1 fibre (S1) was used to generate 12 concrete mixtures with varying lengths of fibre (20 mm, 30 mm, and 40 mm), each length accounting for a different percentage (0.5%, 1%, 1.5%, and 2%) of fibres that replaced the cement in the concrete mix. The test components were submerged in a water tank for 28 days at ambient temperature (20 ± 5 °C). According to the UK Department of the Environment (DoE1975) method for the “Design of Normal Concrete Mixes” [55], the technique adhered to the standard 28-day hardening requirement for concrete. Figure 2 presented the comprehensive schematic for the experimental plan.

### 2.3. Mechanical Tests

Fresh-state and mechanical property tests to observe the impact of FRC, including its compressive, and flexural strengths, were carried out in this study.

#### 2.3.1. Compressive Strength Test

The compressive strength test of cube samples is typically performed in the laboratory by placing concrete cube specimens under a controlled hydraulic pressure machine and utilizing a Universal Hydraulic Test Machine to test the compressive strength at three-time intervals, as shown in Figure 3a. A Universal Hydraulic Test Machine was used to perform the compressive test using load control by a displacement of approximately 5 mm. For each concrete mixture, the average of three cubes for treatment periods (28 curing days) was tested by EN 12390-3:2009 specifications, which was confirmed by the Standard (BS EN 12390-4:2000) [56]. The test for compressive strength is the most critical performance indicator for measuring concrete’s strength. Essential characteristics of concrete include the pressure’s strength and the material’s durability.

#### 2.3.2. Flexural Strength Test

The test for flexural strength is the most critical performance indicator for measuring concrete strength. Essential characteristics of concrete include the bending strength. The flexural strength test of beam samples is typically performed in the laboratory by placing concrete prism specimens under a controlled hydraulic pressure machine and utilizing a Universal Hydraulic Test Machine to test the flexural strength at three-time intervals, as shown in Figure 3b. A Universal Hydraulic Test Machine was used to perform the flexural test using load control by a displacement of approximately 5 mm. The flexural strength test to performed by BS EN 12390-5:2009 and the flexural strength was determined from the average of the three specimens [57].

### 2.4. Pre-Processing the Data for the Empirical Model

#### Multiple Linear Regression Method

Identifying the relationship between two or more variables is a common task in engineering. Using statistical regression aids in forming mathematical equations for observable phenomena, and MLR is widely used to express the relationship between several independent variables and a dependent variable. The dependent variable, which has multiple equivalent coefficients, is determined by the number of parameters [58,59]. When there are more than two independent variables, multiple regression is performed. MLR evaluates the relationship between two or more input variables by adapting a linear equation to the observed data [57,60]. Regression analysis can be used to estimate the relationship between the variables. Modelling and analysing one dependent variable and one or more independent variables are the mainstays of this technique [61]. The goal of regression is to minimize the difference between experimental and predicted results using the principle of least squares [62]. The procedure entails selecting an appropriate initial form for the equation that closely resembles the correlation between the independent and dependent variables. An MLR model was used to explore and compare the effect of the input parameters (i.e., the type, shape, length, and percentage).
(1)$$\hat{Y}=b_{0}+b_{1}X_{1}+b_{2}X_{2}+b_{3}X_{3}+b_{4}X_{4}$$
where, $X_{1}$ indicates the type, $X_{2}$ represents the shape, $X_{3}$ represents the length, $X_{4}$ represents the percentage of fibres, and $b_{0}$ is the predicted regression coefficient that represents the association between the dependent variable $\hat{Y}$ and parameters X. The current work proposes a Khan Khalel model that can predict the desirable compressive and flexural strengths for any given values of key fibre parameters. The Khan Khalel model used four dependent parameters to predict the compressive and flexural strengths of FRC. First, we considered the effect of fibre type on the elastic modulus because the modulus varies according to the fibre. The second parameter is shape, as the value of tensile strength differs for the different fibre shapes for the same type of fibre. The third parameter is the percentage of fibres, whereby four different proportions were used in this study: 0.5%, 1%, 1.5%, and 2%. Finally, the fourth parameter, length, was assessed according to three lengths: 20 mm, 30 mm, and 40 mm as shown in Table 1. To eliminate the influence of variances in properties, such as dimension and order of importance between variables, the input parameters for the FRC were transformed from their original values in Table 1 into standardized dimensionless values. This allowed the effect sizes of different variables to be compared. Matlab© multi linear regression command is used in our paper to generate the relevant coefficients for developing the proposed Khan Khalel model discussed in Section 5.

### 2.5. Methods for Evaluating the Accuracy of the Prediction Model

In general, when evaluating the implementation of a prediction method, it is critical to employ a variety of assessment criteria to determine the performance of the model. In this study, four metrics are used to check the predictive accuracy: MAPE, MSE, and R^2^. The metrics are as follows:Mean absolute percentage error (*MAPE*): this is one of the most common metrics used to measure the forecasting accuracy of a model, as shown in Equation (2). The purpose of the *MAPE* formula is to gauge how different the measured value is from the exact value [63].
(2)MAPE=(1/n)×Σ(|actual−forecast|/|actual|)×100
where, Σ is sum, n is the sample size, *actual* is the actual data value, and *forecast* is the data value forecast.Mean squared error (MSE) is another common metric used to measure the prediction accuracy of a model [64]. MSE is calculated as shown in Equation (3):
(3)$$MSE=(1/n)\times \Sigma (\mathrm{actual}-\mathrm{forecast}{)}^{2}$$
where, Σ is sum, n is the sample size, *actual* is the actual data value, and *forecast* is the data value forecast.

## 3. Identification the Parameters of Numerical Model

### 3.1. Description of the Numerical Model

The cube and beam geometry design procedures of the experiments given in the previous part were created in this study using Abaqus software 2019 [65]. The concrete model consisted of plain concrete (cube, beam), steel and plastic fibres, and loading/support. Embedding the fibres in the concrete region is assumed to lead to a perfect bond between the concrete and the fibres. It is worth mentioning that slipping behaviours have the same bond idea for both beam and cube. Despite this, the perfect bonding assumption has been widely utilized in the literature for concrete-like structures [66,67].

The concrete damage plasticity model (CDPM) is a constitutive model that can be used to predict the behaviour of concrete in the numerical approach. It describes the constitutive behaviour of concrete based on the introduction of scalar damage variables. The four main components of the CDPM are damage evolution, yield criterion, law of hardening/softening, and flow rule. CDPM characterizes the compressive and tensile responses of concrete. The overall strain, ε, can be split into two components according to the standard elastic-plasticity theory to reflect concrete nonlinearity and irreversible deformation, as shown in Equation (4). The CDPM includes a scalar damage variable, d,0≤d≤1, and uniaxial compressive/tensile damage variables, $d_{c}$ and $d_{t}$, for simulating progressive material deterioration, as shown in Equations (5)–(8).
(4)$$\varepsilon =\varepsilon^{el}+\varepsilon^{pl}$$
(5)$$\sigma_{ij}=\left(1-d\right)D_{ijkl}^{el}\left(\varepsilon_{ij}-\varepsilon_{ij}^{pl}\right)$$
(6)$$\sigma_{c}=\left(1-d_{c}\right)E_{0}\left(\varepsilon_{c}-\varepsilon_{c}^{pl}\right)$$
(7)$$\sigma_{t}=\left(1-d_{t}\right)E_{0}\left(\varepsilon_{t}-\varepsilon_{t}^{pl}\right)$$

While it is given for uniaxial cyclic loading conditions as
(8)$$d=1-\left(1-s_{t}d_{c}\right)\left(1-s_{c}d_{t}\right)$$

The yield surface specifies the crucial stress level at which plastic deformation is predicted to begin. Many yield criteria have been proposed to account for strength evolution under tension and compression. The CDPM finally adopted the classic criterion first proposed by Lubliner et al. [68] and then refined by Lee and Fenves [42].
(9)$$F=\frac{1}{1-\alpha }(\bar{q}-3\alpha \bar{p}+\beta (\varepsilon^{pl})\langle {\bar{\sigma }}_{max}\rangle -\gamma \langle -{\bar{\sigma }}_{max}\rangle )-{\bar{\sigma }}_{c}(\varepsilon_{c}^{pl})=0$$
(10)$$\begin{array}{c} \alpha =\frac{\left(\sigma_{b0}/\sigma_{c0}\right)-1}{2\left(\sigma_{b0}/\sigma_{c0}\right)-1};\beta =\frac{\sigma_{c0}\left(\varepsilon_{c}^{pl}\right)}{\sigma_{t0}\left(\varepsilon_{t}^{pl}\right)}(1-\alpha )-(1+\alpha )\\ \gamma =\frac{3\left(1-K_{c}\right)}{2K_{c}-1} \end{array}$$

The hardening law describes the pre-peak behaviour when the elastic area ends, whereas the softening law covers the post-peak behaviour throughout the plastic flow [69]. Anisotropic hardening is considered in Abaqus, as shown in the analogous plastic drives as well as the strain evolution law, as shown in Equations (9) and (10).
(11)$$\varepsilon_{c}^{in}=\varepsilon_{c}-\sigma_{c}/E_{0}$$
(12)$$\varepsilon_{t}^{ck}=\varepsilon_{t}-\sigma_{t}/E_{0}$$

The compressive and tensile inelastic strains are $\varepsilon_{c}^{in}$ and $\varepsilon_{t}^{ck}$, respectively. Plastic deformation is determined by the flow rule, which is guided by a potential flow function as shown in Equations (11) and (12). The CDPM uses a non-associated possible flow rule due to the variations between metal and non-metal materials, and the possible function, *G*, has a hyperbolic Drucker-Prager type form, as shown in Equations (13)–(15):(13)$$\begin{array}{c} \varepsilon_{c}^{pl}=\varepsilon_{c}^{in}-\frac{d_{c}}{\left(1-d_{c}\right)}\frac{\sigma_{c}}{E_{0}} \end{array}$$
(14)$$\begin{array}{c} \varepsilon_{t}^{pl}=\varepsilon_{t}^{ck}-\frac{d_{t}}{\left(1-d_{t}\right)}\frac{\sigma_{t}}{E_{0}} \end{array}$$
(15)$$G=\sqrt{{\left(e\sigma_{t0}tan\;\psi \right)}^{2}+{\bar{q}}^{2}}-\bar{p}tan\;\psi =0$$

The CDPM in Abaqus is utilized in this study to describe compressive strength in normal concrete, assuming that the fibres are randomly distributed in the matrix and the FRC is thus considered as a homogeneous material. The default settings of the model parameters defining its operation in a complex stress state (ψ, f, e, *Kc*) [70] were used for the numerical analysis of FRC beams and cubes, and are shown in Table 3. According to the standard [71], the Poisson’s ratio of uncracked concrete is supposed to be 0.2. Table 4 and Table 5 present a representative summary of the concrete characteristics of the Abaqus software, as described by Shin et al. [72]. The effects on FRC were carefully monitored and minimized in this research using concrete damage plasticity DCP models to obtain quasi-static behaviours from the Standard-Explicit model simulation. Readers are urged to look up comprehensive discussions on quasi-static simulations using the Standard-Explicit model in the literature [73,74,75]. A fixed boundary condition (BC) was used at the bottom of the cube surface, with vertical displacement defined at the top. The BCs of the experiment were simplified by employing a 50 mm diameter supporting and loading cell, as shown in Figure 4. A fixed BC was used for the beam sample at the bottom of the supporting cell, with vertical displacement defined at the top of the loading bars. As the finite element (FE) simulations were run using the Standard-Explicit model, the vertical displacement was applied smoothly and slowly to minimize any noticeable load effects. The function “*AMPLITUDE, DEFINITION = SMOOTH STEP” was utilized to enforce the smooth and slow displacement BC. The bulk viscosity option was chosen to limit the load effects on the numerical results [65]. The constitutive plasticity rule was used to model the nonlinear, elastic behaviour (containing four different fibre shapes) of the steel and plastic fibres. The plasticity model assumed that steel and plastic fibres behaved similarly to the steel reinforcement behaviour used in the literature [66,76,77]. Digimat-FE software 2019 was used for the composite materials to obtain the random distribution of fibres inside the concrete samples [78]. The fibre distribution and interaction between fibres and concrete are shown in Figure 5. The small yellow square is the interaction between load and support cells with the surface, and the small yellow circle is the interaction between fibre and concrete as shown in Figure 5. Tensile tests, comprising elastic modulus, stress, and strain, were used to provide the inputs for the plasticity model. The plasticity properties of the fibres are shown in Table 6 and Table 7.

### 3.2. Concrete Mesh Convergence Analysis

The process of mesh convergence entails reducing the element size and analyzing the effect of this reduction on the solution’s precision. The smaller the mesh size, the more precise the solution, as the behaviour of the design or product is sampled more precisely across its physical domain. The accuracy of numerical results is generally highly dependent on the mesh size utilized in the numerical model. More accurate results can be produced using a smaller mesh size, but this is more computationally expensive and requires greater computer capacity. The smallest mesh dimension is not viable for the CDPM due to software and computer limitations. As a result, performing a mesh convergence study to identify the ideal mesh size is critical. Four concrete cubes with varying mesh sizes (10 mm, 8 mm, 6 mm, 4 mm, and 3 mm) were utilized in the CDPM for the mesh convergence analysis to establish the ideal element size of the concrete model. The conductivity signatures derived from the four concrete cubes are compared in Figure 6. The results reveal a considerable difference in conductance mesh size between the 10 mm, 8 mm, and 6 mm elements. However, the difference between the 4 mm and 3 mm elements is relatively modest. In this study, a mesh size of 5 mm was employed to represent the concrete cube for concrete disaster response assessments, when considering process time and computer memory. The same mesh convergence analysis steps were performed for the four beam sizes for flexural strength, as shown in Figure 7.

## 4. Results and Discussion

### 4.1. Influence of Fibres Parameters on Compressive Strength

As shown in Figure 8, the compressive strength of the control concrete was greater than that of the fibre-containing concrete. The results indicate that steel and plastic fibres were added, resulting in a weak composite with low compressive strength. However, flexural strength properties were increased. The results indicate that the compressive strength of concrete increased slightly for two lengths of PFRC2 fibre in the concrete mix (20 mm and 30 mm). However, as indicated in Figure 8, there were no compressive impacts over a length of 40 mm. The compressive strength dropped slightly when the percentage of SFRC1 was increased by adding fibres of 20 mm, 30 mm, and 40 mm to the concrete mix. In contrast, this slight compression was higher than the concrete mixture’s planned compression level of 30 MPa. The compressive strength increases as the proportion of SFRC2 at 40 mm in the concrete mix increases. However, at lengths of 20 mm and 30 mm, no compressive influence was seen. In past, researchers tested the compressive strength of the FRC and discovered that the fibres enhanced stress resistance. Similarly, other researchers [79,80] substituted coarse and fine particles with fibres to improve the mechanical properties of concrete. Other research suggested that the use of steel and e-plastic reduces compressive strength because the aggregate is replaced by steel and plastic fibres [81,82,83,84,85,86].

As shown in Figure 8, the compressive strength of concrete has a minimum and max standard deviation between 0.17 and 2.5 MPa. It shows that the data points are tightly clustered around the mean value. This means that there is not much difference between the compressive strengths of the concrete samples in the collection. Together, these two numbers tell us a lot about the compressive strength of the components of concrete in the dataset. The low standard deviation shows that the samples of concrete are all about the same, and the mixing shows the strength of all samples on average.

**Figure 8 materials-16-03700-f008:**
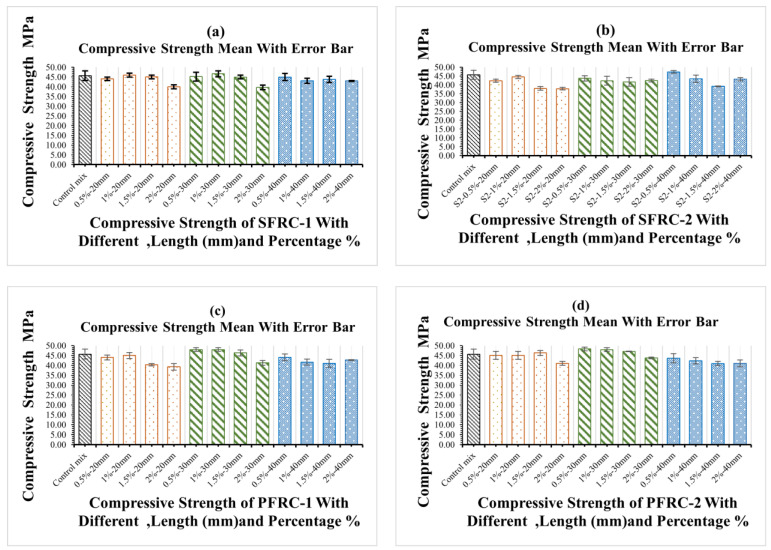
Impact of different types of fibres on concrete compressive strength (**a**) compressive strength of PFRC-1, (**b**) compressive strength of PFRC-2, (**c**) compressive strength of SFRC-1 and (**d**) compressive strength of SFRC-2.

Overall, the compressive strength of FRC decreased when compared to the compressive strength of standard concrete. Moreover, fibre length has a negative effect on the compressive qualities when considering how the fibre content influences them, significantly when the length is increased, as evidenced by the results, which showed a drop in compressive strength.

### 4.2. Influence of Fibres Parameters on Flexural Strength

A flexural load was applied to 98 prisms of FRC samples with various lengths, forms, and fibre volume percentages. Figure 9 illustrates the influence of fibre type, shape, and percentage on flexural strength after 28 days for different fibre types. The flexural strength increased somewhat when the percentage of plastic-1 fibre in the concrete mix with lengths of 20 mm, 30 mm, and 40 mm increased. Compared to conventional concrete, Plastic-2 FRC showed excellent ductility for all used fibre lengths and percentages. The flexural strength increased significantly with the steel fibre percentage for steel-1 in the concrete mixture. All mixture samples with various fibres have higher flexural strength than the control concrete. Furthermore, when the amount of fibre increases, so does the transverse deformation of the sample. The stress induced by the external load is effectively communicated between the steel fibre and concrete matrix, allowing the steel and plastic fibres to completely utilize their flexural strength and compensate for the FRC matrix’s lower tensile capacity. Steel fibre can bridge large cracks because the interface between steel, plastic fibre, and concrete has a specific bond strength. Therefore, the specimen will not have a severe fracture because the cracks are constantly growing. However, the number of cracks will increase significantly during this procedure, as will the crack width. As a result, the crack expands dramatically, causing the transverse deformation of the specimen to increase. Previous research findings have validated the technique of enhancing flexural strength by increasing polypropylene fibre content, as it demonstrated improved flexural performance in both equivalent durability and flexural strength with an increase in the percentage of SF used [87,88]. The increase in flexural strength is much more significant than the increase in compressive strength. As a result, the impact of fibre content on flexural characteristics is substantial when the length of the fibre is increased as shown in Figure 9. The more significant the reduction in cement usage, the greater the reduction in eCO_2_ and the further the realization of sustainable growth in the construction industry.

As shown in Figure 9, the flexural strength of concrete has a minimum and max standard deviation between 0.04 and 0.27 MPa. It shows that the data points are tightly clustered around the mean value. This means that there is not much difference between the flexural strengths of the concrete samples in the collection. Together, these two numbers tell us a lot about the flexural strength of the components of concrete in the dataset. The low standard deviation shows that the samples of concrete are all about the same, and the mixing shows the flexural strength of all samples on average.

**Figure 9 materials-16-03700-f009:**
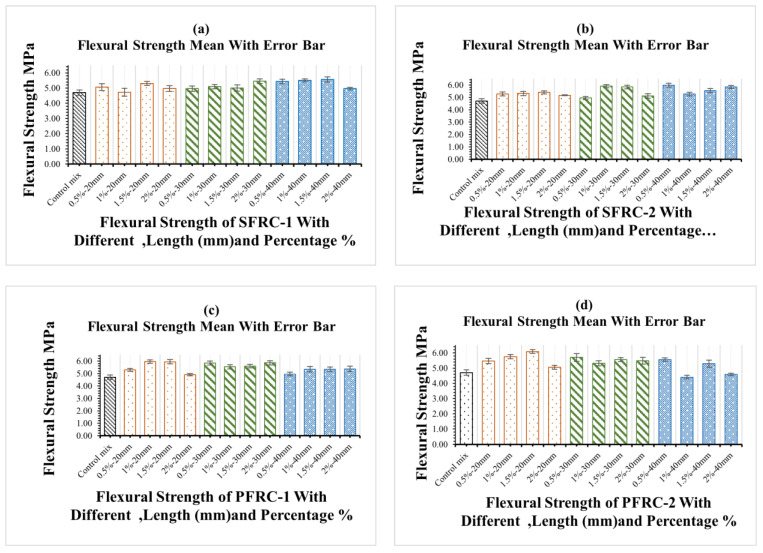
Impact of different types of fibres on concrete flexural strength. Impact of different types of fibres on concrete flexural strength (**a**) flexural strength of PFRC-1, (**b**) flexural strength of PFRC-2, (**c**) flexural strength of SFRC-1 and (**d**) flexural strength of SFRC-2.

In general, FRC’s flexural strength increases compared to standard concrete’s flexural strength. Moreover, fibre length negatively influences the flexural strength when considering how the fibre content affects them, significantly when the length is increased, as evidenced by the results, which showed a decrease in flexural strength.

### 4.3. Numerical Model

#### 4.3.1. Numerical Analysis of Flexural Strength

In numerical model, the samples were notched in the centre and failed in their flexural capacity, with a substantial crack down the length of the beam. The typical failure mechanisms of the 48 beams are shown in Figure 10 and Figure 11. Although the beam geometry, boundary, and loading conditions were all symmetric, the cracks in several of the beams were slightly convoluted and strayed from the centre beamlines. This is due to the random distribution and orientation of the steel and plastic fibres [89,90], which increases the crack-tip stress fields and the extremely heterogeneous local tensile strength and fracture toughness, as in ordinary concrete where aggregates behave similarly to steel fibres [91]. This cannot be accurately represented by the homogeneous models used in this investigation and can only be simulated by models that directly or indirectly consider random heterogeneity [22,92,93]. According to the CDPM, two failure processes for flexural strength were observed in this study. When a vertical crack propagates from the corners of the opening to the applied load and support, the first mode is a diagonal splitting failure. When a diagonal fracture forms in the shear span, it causes high strains inside the compression chord of the apertures next to the site of the loads. Figure 10 shows how the damage to concrete plasticity in the simulation creates failures consistent with actual observations. There were noticeable differences between the numerical and experimental data for the mechanical properties in previous research, including flexural and cracking tensile strength [94,95,96]. The-max load in the middle and max displacement of SFRC-2 beams obtained from the numerical model, and that load was used to calculate the flexural strength according to BS EN 12390-6 (2009). This is comparable to the experimental results obtained from the specimens in this study, as shown in Table 1 with different lengths, shapes, and percentages of fibre. The contour plots in Figure 10 and show the damage levels in various colours, ranging from the most severe to the least, in red, yellow, green, and blue. The images show that the crack propagation of beam distribution is equal along the middle axis, similar to the beam in the experimental observation. The estimation of the flexural strength for all the beams is also acceptable in terms of agreement with the experimental results.

#### 4.3.2. Numerical Analysis of Compressive Strength

The CDPM output can provide information on the damage distribution and characteristics of the concrete cube at various stages through numerical analysis. The concordance between the estimates from the suggested formula and the findings of the numerical analysis demonstrates compressive strength [97,98]. The max load applied on the cube of SFRC-1 at 1% and max displacement obtained from the numerical model, and that load was used to calculate the compressive strength according to BS EN 12390-4 (2000). This is comparable with the experimental results obtained from the specimens, as shown in Figure 12b. Figure 12c–e shows the damage contours of the three views (YX, ZX, and YZ) in Figure 12a. The contour plots show damage levels in various colours, ranging from the most severe to the least, in red, yellow, green, and blue. The images show that the damage distribution is equal along the middle axis, like the experimental observation. Furthermore, by observing the damage shapes derived by CDP and testing, it is discovered that the top and bottom surfaces of the concrete cubes stay relatively intact. Figure 12c,d shows that the damage modes at the YX and ZX portions of the specimen are X-shaped (d). The concrete mixture peeled off on four edges in all three concrete cube tests, and the samples were in a dumbbell shape at the end of the test. As a result, the numerically simulated damage contours of the sample correlate well with the experimental observations. The CDPM can be used in the compressive strength check and damage evaluation.

### 4.4. Comparison between the Numerical Model and Experimental Results

#### 4.4.1. Compressive Strength

Compressive strength is one of concrete’s most important mechanical qualities when designing a structure. The Figure 13 shown the comparison of steel and plastic fibre reinforced concrete influence on compressive strength between experimental results and numerical simulations. The compressive strength of the samples ranged from 39.17 to 48.4 MPa, as indicated in Table 8. According to Figure 13, the compressive strength agrees well with the CDPM outcomes. The CPD model in this study might be an effective tool to validate the compressive strength of FRC, rather than conducting laboratory experiments based on material properties such as mechanical properties, length, and fibre percentage of fibre, concrete damage plasticity (CDP) behaviour. The CDP model has the highest accuracy in forecasting compressive strength, with a determination coefficient has good bond. According to the MAPE-evaluated tool results, only four of the numerical values in the CDP model showed an error (9.61%, 8.34, 8.67, 6.84%) between all experimental concrete strength output values (48 experimental findings). 

As demonstrated in this study, the CPD model can predict the optimal fibre with acceptable accuracy to obtain a compressive strength adequate for usage in a structure. This may stimulate the use of the concrete damage plasticity model in forecasting the compressive properties of concrete generated by different fibre materials in the future. To assess the accuracy of the numerical model in this study, the compressive strength of SFRC had the lowest and most significant errors, and the MAPE was between 0.52% and 5.53%; Figure 14 shows the MAPE results for the experimental and numerical models. Table 8 shows the evaluations used to analyses the results output of compressive strength. As shown in Figure 15 the selected linear polynomial seems a poor fit, but it is fine to predict the values of compressive s as the standard deviation is within a range of (0.03 to 2.6) and (0.04 to 0.8) MPa.

**Figure 13 materials-16-03700-f013:**
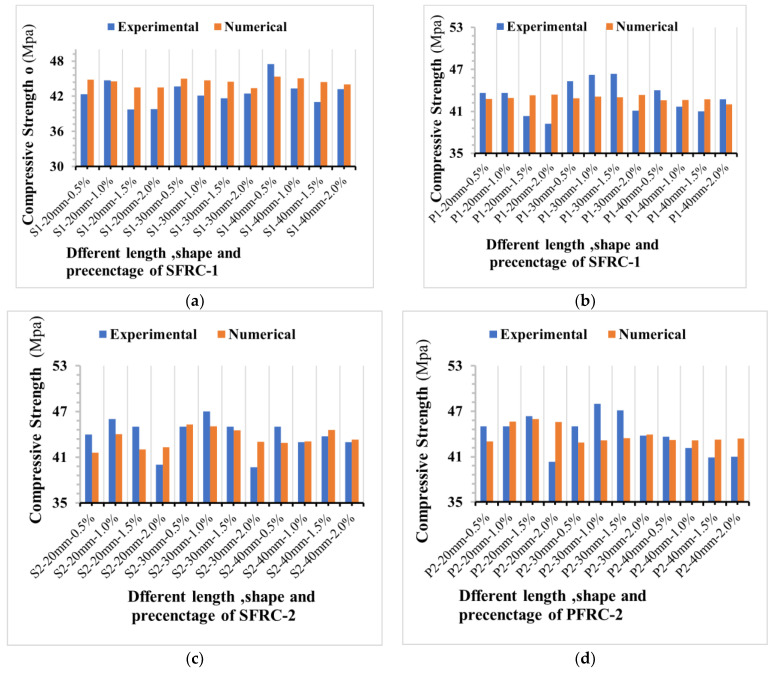
Comparison of steel and plastic fibre reinforced concrete influence on compressive strength between experimental results and numerical simulations. (**a**) the compressive strength of SFRC-1, (**b**) the compressive strength of SFRC-2, (**c**) the compressive strength of PFRC-1 and (**d**) the compressive strength of PFRC-2.

**Table 8 materials-16-03700-t008:** Shows evaluates the accuracy of predicting the compressive strength of fibre-reinforced concrete.

Mixes of Concrete	Elastic Modulus of Fibre Types (MPa)	Tensile Strength of Different Type Shape (MPa)	Fibres Length (mm)	Percentage of Fibre (%)	Compressive Strength Experimental	Compressive Strength Numerical (MPa)	(MSE) Numerical and Experimental	(MAPE) Numerical and Experimental
					(MPa)			
SFRC-1	200,000	800	20	0.5	42.33	44.53	1.48	4.94
SFRC-1	200,000	800	20	1	44.7	45.84	1.07	2.49
SFRC-1	200,000	800	20	1.5	39.73	43.50	1.94	8.67
SFRC-1	200,000	800	20	2	39.8	42.50	1.64	6.35
SFRC-1	200,000	800	30	0.5	43.67	45.02	1.16	3.00
SFRC-1	200,000	800	30	1	42.13	44.00	1.37	4.25
SFRC-1	200,000	800	30	1.5	41.63	44.00	1.54	5.39
SFRC-1	200,000	800	30	2	42.47	43.36	0.94	2.05
SFRC-1	200,000	800	40	0.5	47.47	46.34	1.06	−2.44
SFRC-1	200,000	800	40	1	43.33	44.40	1.03	2.41
SFRC-1	200,000	800	40	1.5	41	43.40	1.55	5.53
SFRC-1	200,000	800	40	2	43.2	42.00	1.10	−2.86
PFRC-1	7000	465	20	0.5	43.63	45.00	1.17	3.04
PFRC-1	7000	465	20	1	43.63	44.91	1.13	2.85
PFRC-1	7000	465	20	1.5	40.33	43.29	1.72	6.84
PFRC-1	7000	465	20	2	39.24	43.41	2.04	9.61
PFRC-1	7000	465	30	0.5	45.3	46.87	1.25	3.35
PFRC-1	7000	465	30	1	46.23	47.09	0.93	1.83
PFRC-1	7000	465	30	1.5	46.37	46.00	0.61	−0.80
PFRC-1	7000	465	30	2	41.1	43.37	1.51	5.23
PFRC-1	7000	465	40	0.5	44	44.60	0.77	1.35
PFRC-1	7000	465	40	1	41.67	42.62	0.97	2.23
PFRC-1	7000	465	40	1.5	41	42.71	1.31	4.00
PFRC-1	7000	465	40	2	42.7	43.00	0.55	0.70
PFRC-2	7000	552	20	0.5	45	46.04	1.02	2.26
PFRC-2	7000	552	20	1	45	45.60	0.77	1.32
PFRC-2	7000	552	20	1.5	46.33	46.00	0.57	−0.72
PFRC-2	7000	552	20	2	40.33	44.00	1.92	8.34
PFRC-2	7000	552	30	0.5	45.01	45.90	0.94	1.94
PFRC-2	7000	552	30	1	48	45.17	1.68	−6.27
PFRC-2	7000	552	30	1.5	47.1	47.45	0.59	0.74
PFRC-2	7000	552	30	2	43.77	44.00	0.48	0.52
PFRC-2	7000	552	40	0.5	43.67	44.20	0.73	1.20
PFRC-2	7000	552	40	1	42.17	43.15	0.99	2.27
PFRC-2	7000	552	40	1.5	40.9	42.20	1.14	3.08
PFRC-2	7000	552	40	2	41	43.42	1.56	5.57
SFRC-2	200,000	1150	20	0.5	44	44.58	0.76	1.30
SFRC-2	200,000	1150	20	1	46	45.04	0.98	−2.13
SFRC-2	200,000	1150	20	1.5	45	45.50	0.71	1.10
SFRC-2	200,000	1150	20	2	40	41.30	1.14	3.15
SFRC-2	200,000	1150	30	0.5	45	46.26	1.12	2.72
SFRC-2	200,000	1150	30	1	47	47.04	0.20	0.09
SFRC-2	200,000	1150	30	1.5	45	45.54	0.73	1.19
SFRC-2	200,000	1150	30	2	39.67	42.05	1.54	5.66
SFRC-2	200,000	1150	40	0.5	45	45.00	0.00	0.00
SFRC-2	200,000	1150	40	1	43	43.08	0.28	0.19
SFRC-2	200,000	1150	40	1.5	43.77	44.58	0.90	1.82
SFRC-2	200,000	1150	40	2	43	43.33	0.57	0.76

**Figure 14 materials-16-03700-f014:**
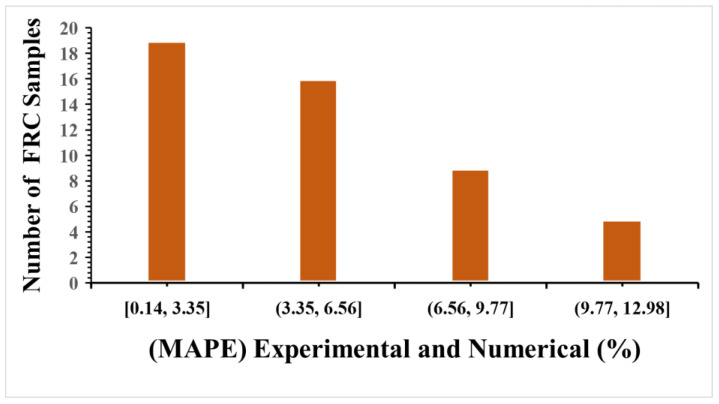
Mean absolute percentage error (MAPE) of compressive strength results.

**Figure 15 materials-16-03700-f015:**
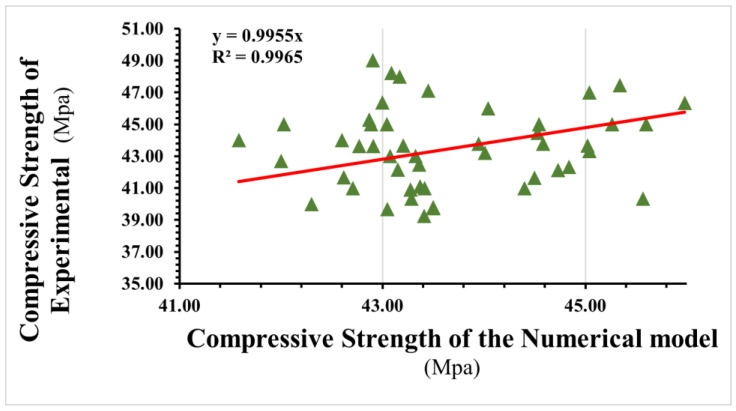
Relationship between compressive strength results for the experimental and numerical models.

#### 4.4.2. Flexural Strength

As shown in Table 9, the experimentally determined mean values for flexural strength fall agree with the value of flexural strength obtained by the numerical models. Figure 16 shows the comparison of steel and plastic fibre reinforced concrete influence on compressive strength between experimental results and numerical simulations.

The match between the numerical data and experimental outcomes is significantly high as shown in Figure 16. This enables the prediction of the flexural strength of fibre-reinforced concrete using the Abaqus software’s concrete damage plasticity.

**Figure 16 materials-16-03700-f016:**
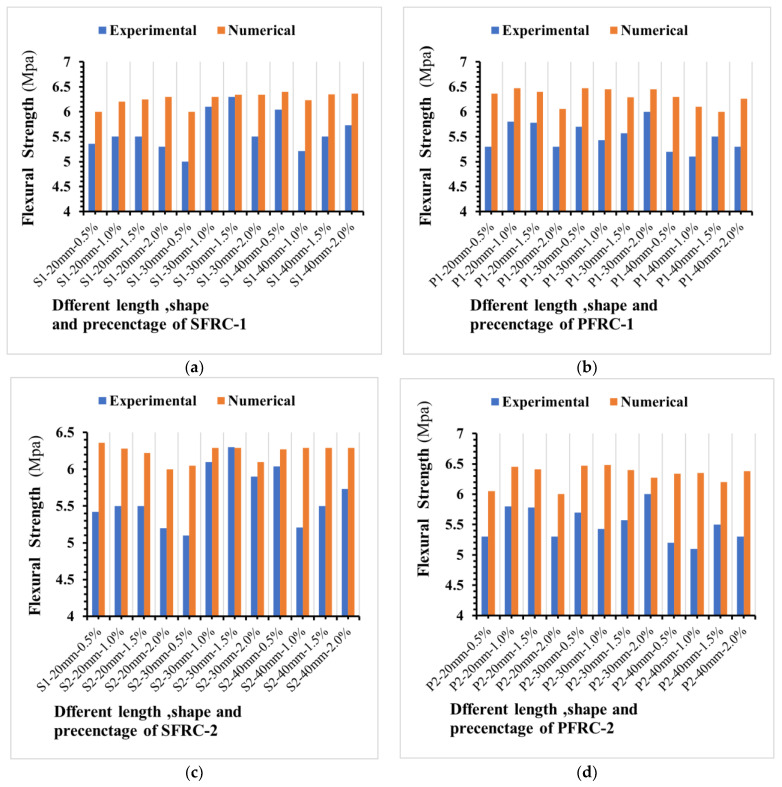
Comparison of steel and plastic fibre reinforced concrete influence on flexural strength between experimental results and numerical simulations. (**a**) the flexural strength of SFRC-1, (**b**) the flexural strength of SFRC-2, (**c**) the flexural strength of PFRC-1 and (**d**) the flexural strength of PFRC-2.

As shown in Figure 17, the selected linear polynomial seems a poor fit, but it is fine to predict the values of flexural strength as the standard deviation is within a range of (0.04 to 0.8) MPa.

According to the MAPE evaluated tool results, only four of the numerical values in the CDP model showed an error (16.67%, 15.70, 14.75%, 14.59%) between all experimental concrete flexural strength output values (48 experimental findings). As demonstrated in this study, the CPD model can predict the optimal fibre with acceptable accuracy to obtain a flexural strength adequate for using in a structure. This may stimulate the use of the concrete damage plasticity model in forecasting the flexural strength properties of concrete generated by different fibre materials in the future. To assess the accuracy of the numerical model in this study, the flexural strength of SFRC had the lowest and most significant errors, and the MSE was between 0.32% and 1.03%. Figure 18 shows the MAPE results for the experimental and numerical models. 

**Figure 17 materials-16-03700-f017:**
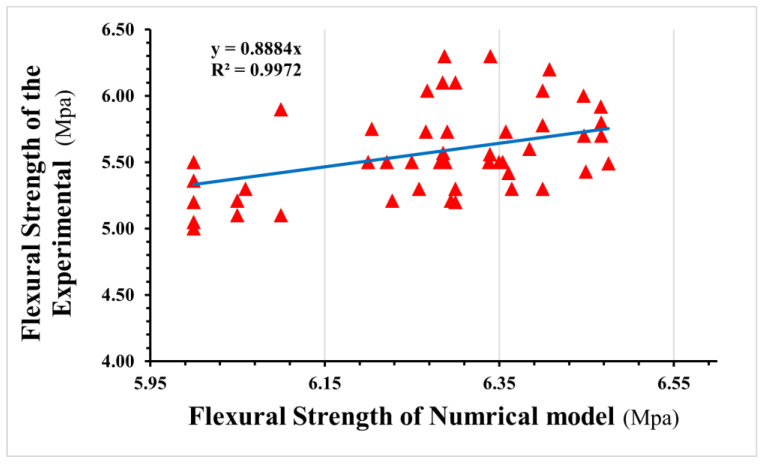
Relationship between flexural strength results for the experimental and numerical models.

This simple model for evaluating the behaviour of FRP-reinforced concrete could result in more engineers utilising this concrete form in actual applications. These results indicated that the numerical model had an adequate predictive method between input materials and output attributes. 

## 5. Development of Khan Khalel Model

The compressive and flexural strength values on various combinations of fibre input parameters were mentioned above. These results are presented in Table 8 and Table 9. An empirical model has been developed based on the discussed experimental values which will help to estimate FRC properties based on material parameters such as elastic modulus, tensile strength, fibre length, and fibre percentage. MLR as discussed in Section 2.4 is used to develop the model that can examine the effect of fibre characteristics on compressive and flexural strength as shown in Equations (16) and (17). The Khan-Khalel model indicates that an increase in the length and percentage of fibre reduces the overall value of compressive strength.
(16)$$\begin{array}{l} \hat{C.S}=40.72-(1.46\times 10-05)\;Elastic\;modulu\;of\;fiber\;(MPa)\\ \;\;\;\;\;\;+(4.38\times 10-03)\;Tensile\;strength\;of\;fiber\;(MPa)\\ \;\;\;\;\;\;+(0.000604)\;Length\;of\;fiber\;(mm)\\ \;\;\;\;\;\;+(45.21)\;Percentage\;of\;fiber\;(\%) \end{array}$$
where $\hat{C.S}$ is compressive strength of fiber reinforced concrete.

The developed model indicates that an increase in the length and percentage of fibre can increase the overall flexural strength.
(17)$$\begin{array}{l} \hat{F.S}=5.3-\left(5.35\times 10-08\right)\;elastic\;modulus\;of\;fiber\;\left(MPa\right)\\ \;\;\;\;\;\;+\;\left(1.61\times 10+03\right)\;Tensile\;strength\;of\;fiber\;\left(Mpa\right)\\ \;\;\;\;\;\;+\left(0.002656\right)\;Length\;of\;fiber\;\left(mm\right)\\ \;\;\;\;\;\;+\;\left(5.88333\right)\;Percentage\;of\;fiber\;\left(\%\right)\;\% \end{array}$$
where $\hat{F.S}$ is the flexural strength of fiber reinforced concrete.

### Validation of Khan Khalel Model with Numerical Results

The developed empirical model is validated on the arbitrary input values to observe the accuracy for predicting the compressive and flexural strengths of FRC in general with the help of numerical model. We validated the model on same material types and shapes but on different lengths and percentages of fibres. The selected values were 25 mm, 35 mm, 1.25% and 1.75%, respectively. The obtained values of predicted compressive and flexural strengths by Khan Khalel model were compared with numerical estimations as provided in Table 8 and Table 9. We have found a good agreement in between Khan Khalel and numerical models. Additionally, as shown in Table 10 and Table 11, mean absolute error and mean square error were used to validate the results of flexural and compressive strengths. 

The MSE results of the Khan Khalel and numerical model showed an error in the range of (3.84–12%) between all flexural and compressive concrete strengths. The error in estimation of compressive strength is under 10% and hence it determines the goodness of prediction of the proposed model.

**Table 10 materials-16-03700-t010:** Shows the validation of compressive strength.

Mixes of Concrete	Elastic Modulus of Fibre Types (MPa)	Tensile Strength of Different Fibres Shape (MPa)	Fibres Length (mm)	Percentage of Fibre %	Compressive Strength Khan Khalel (MPa)	Compressive Strength Numerical (MPa)	(MSE) Numerical and Khan Khalel	(MAPE) Numerical and Khan Khalel
SFRC-1	200,000	1150	25	1.25	43.75	47.67	15	8.93
SFRC-2	200,000	800	25	1.25	41.98	45.90	15	8.75
PFRC-1	70,000	465	25	1.25	42.31	45.80	12	7.61
PFRC-2	7000	552	25	1.25	43.62	45.70	4	4.56
SFRC-1	200,000	1150	35	1.75	43.5	47.80	14	8.69
SFRC-2	200,000	800	35	1.75	42.11	45.27	10	6.97
PFRC-1	7000	465	35	1.75	43.47	45.70	5	4.89
PFRC-2	7000	552	35	1.75	43.85	45.60	3	3.84

The MSE results of the Khan Khalel values and numerical model showed an error in the range of (0.36–0.86%) between all flexural concrete strength output of parameters that were not used in the experimental. The error MAPE in estimation of compressive strength is under 10% and hence it determines the goodness of prediction of the proposed model.

**Table 11 materials-16-03700-t011:** Shows validation of flexural strength.

Mixes of Concrete	Elastic Modulus of Fibre Types (MPa)	Tensile Strength of Different Type Shape (MPa)	Fibres Length (mm)	Percentage of Fibres (%)	Flexural Strength Khan Khalel (MPa)	Flexural Strength Numerical (MPa)	(MSE) Numerical and Khan Khalel	(MAPE) Numerical and Khan Khalel
SFRC-1	200,000	1150	25	0.0125	5.62	6.49	0.76	13.42
SFRC-2	200,000	800	25	0.0125	5.56	6.49	0.86	14.29
PFRC-1	70,000	465	25	0.0125	5.52	6.22	0.49	11.24
PFRC-2	7000	552	25	0.0125	5.53	6.30	0.59	12.20
SFRC-1	200,000	1150	35	0.0175	5.68	6.28	0.36	9.61
SFRC-2	200,000	800	35	0.0175	5.62	6.49	0.76	13.40
PFRC-1	7000	465	35	0.0175	5.58	6.30	0.53	11.54
PFRC-2	7000	552	35	0.0175	5.59	6.28	0.47	10.97

As demonstrated in this study, the Khan Khalel model can predict the optimal fibre with acceptable accuracy to obtain a compressive strength adequate for usage in a structure. This straightforward method for estimating the behaviour of concrete incorporating FRP could lead to more engineers employing this form of concrete in actual applications. Compared with previous studies, the collection of investigations reveals fascinating conclusions, notably that variation in the stated test results has an accuracy, in some cases ranging from 1–10%. Moreover, This is due to extensive diversity in test specimens, materials, loading configurations, experimental methodologies, and test arrangements [99,100].

## 6. Conclusions

This paper proposed the Khan Khalel model to predict the optimal fibre reinforcement in concrete. The proposed model can take key fibre parameters as inputs and predict the compressive and the flexural strengths of reinforced concrete as a result. The findings of this investigation are as follows:The proposed model can facilitate the users in the construction industry to select an optimal set of fibre properties during reinforcement. The model can be used to predict concrete behaviour with elastic fibre properties and any given physical shape and dimensions;The given results show a good agreement with a numerical model where error represents the challenges in reinforcement such as ideal mixing and distribution of fibres in concrete, difficulty in finding the interfacial properties of fibres with concrete constituents and difficulty in finding the plastic behaviour on compressive and flexural testing machines;Compared to previous studies, interesting results are obtained. The overall variation in the test results ranges from 3.84 to 12%. It is quite acceptable, especially in the presence of extensive diversity in test specimens, materials, loading configurations, experimental methodologies, and test arrangements [98,99];The empirical model prediction accuracy is measured with R^2^ = 0.997, MSE = 8.21, and MAPE = 5.93%, and can be used as a compressive strength prediction tool for FRC. In regard to flexural strength, existing literature models predict with a prediction accuracy measured as R^2^ ranging from 0.78 to 0.87 and MAPE ranging from 6.15 to 17.9 [100]. However, our presented model can more accurately predict flexural strength and its accuracy is measured as R^2^ = 0.996, MSE = 0.64 and MAPE = 12%;The proposed model has used elastic modulus for material input selection and hence predominantly considers linear elastic behaviour in FRC during tests. However, a use of plasticity correction on this input can represent the complex nonlinear relationship between the various input variables and properties of FRC. The different shapes and more dimensions of the fibre should also be considered. This will be considered as a change to be implemented in future work.

## Figures and Tables

**Figure 1 materials-16-03700-f001:**
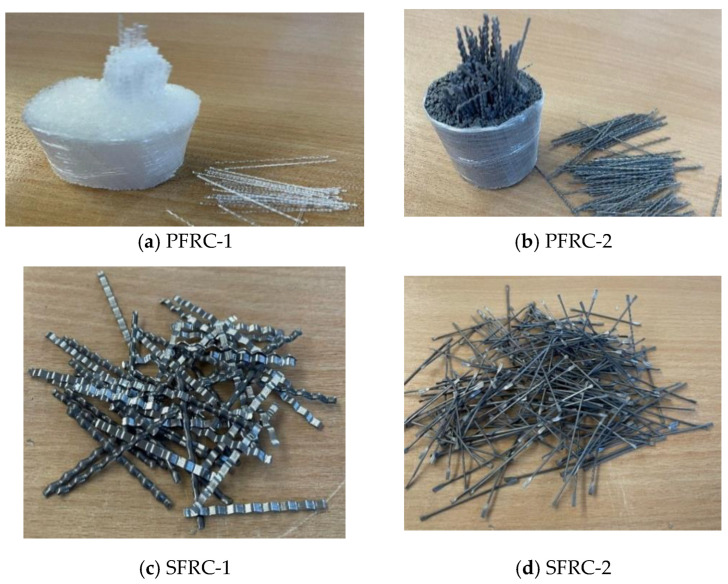
Shape of steel and plastic fibres used in this study: (**a**) macro/monofilament, (**b**) crimped; (**c**) indented and (**d**) dumbbell.

**Figure 2 materials-16-03700-f002:**
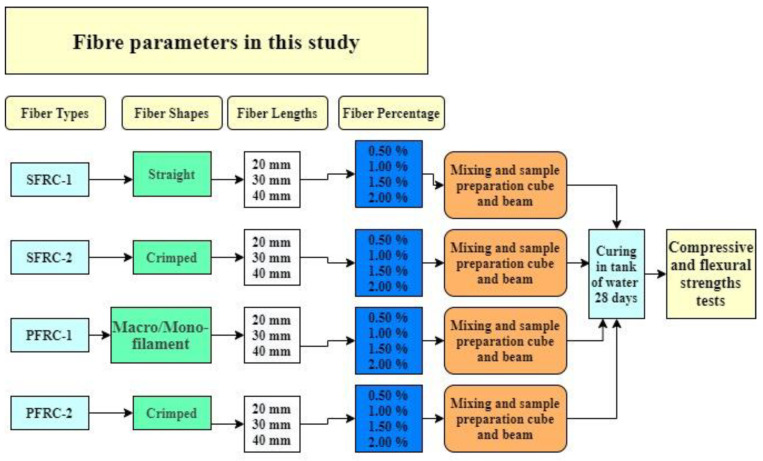
Experimental scheme.

**Figure 3 materials-16-03700-f003:**
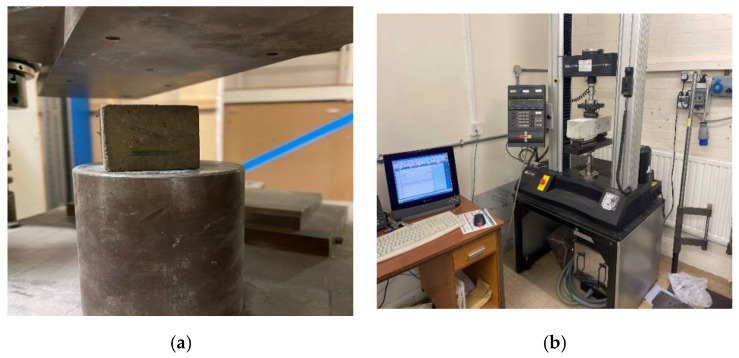
Mechanical tests. (**a**) Compressive strength test. (**b**) Flexural strength test.

**Figure 4 materials-16-03700-f004:**
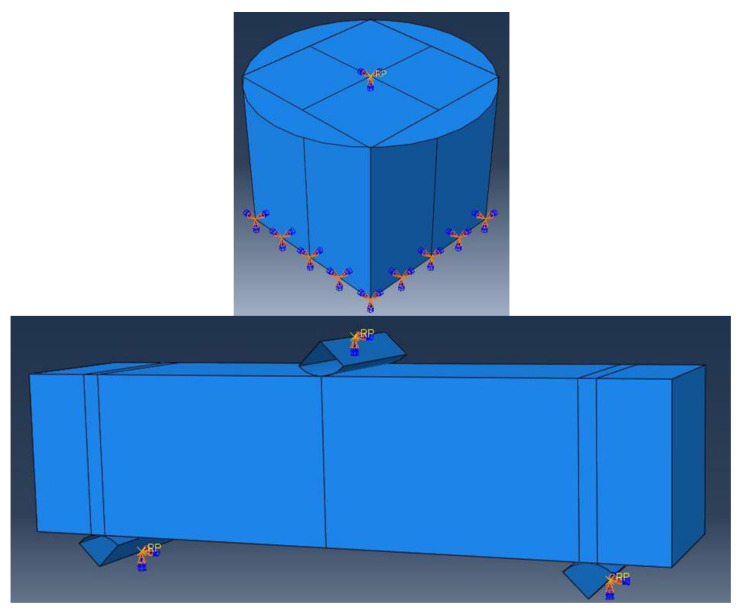
Boundary conditions of the beam and cube.

**Figure 5 materials-16-03700-f005:**
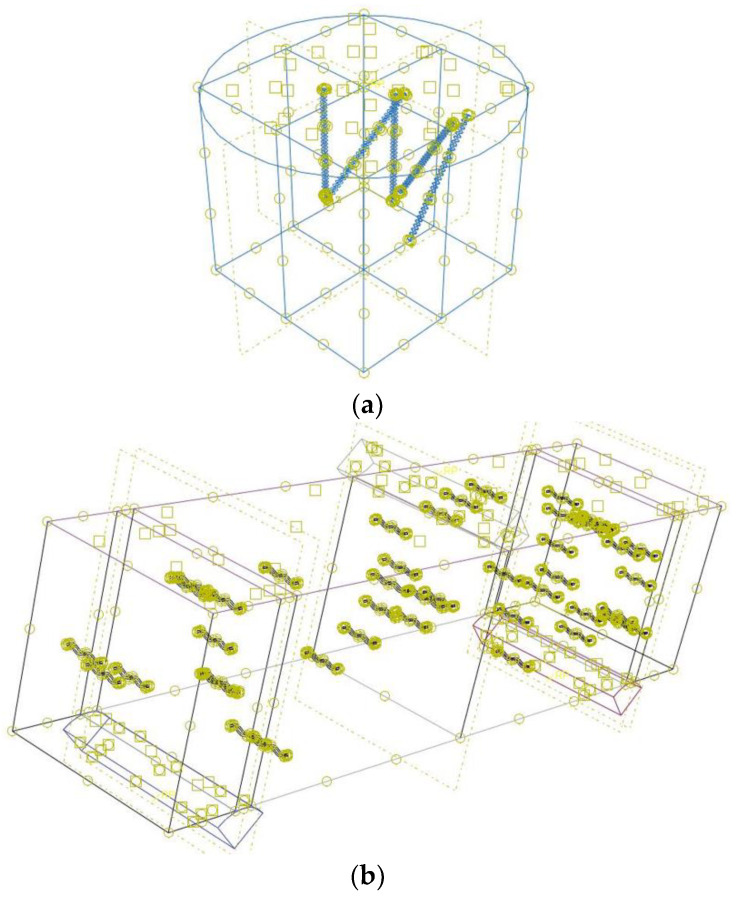
Distribution and interaction between fibres and concrete. (**a**) Cube. (**b**) Prism.

**Figure 6 materials-16-03700-f006:**
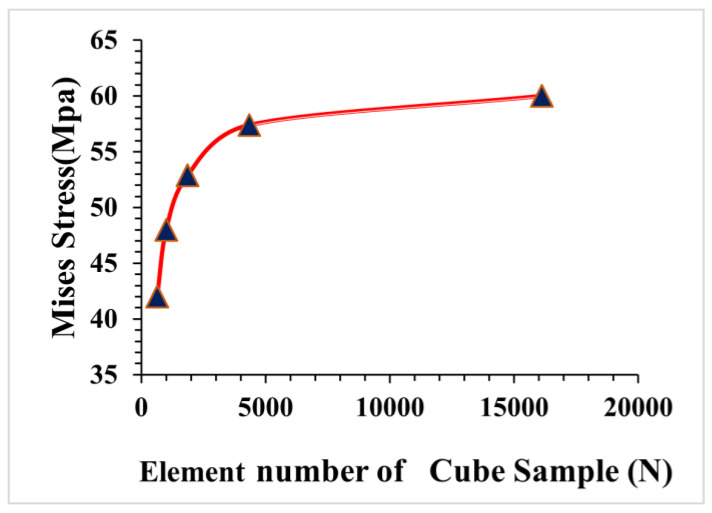
Mesh convergence analysis of FRC cube.

**Figure 7 materials-16-03700-f007:**
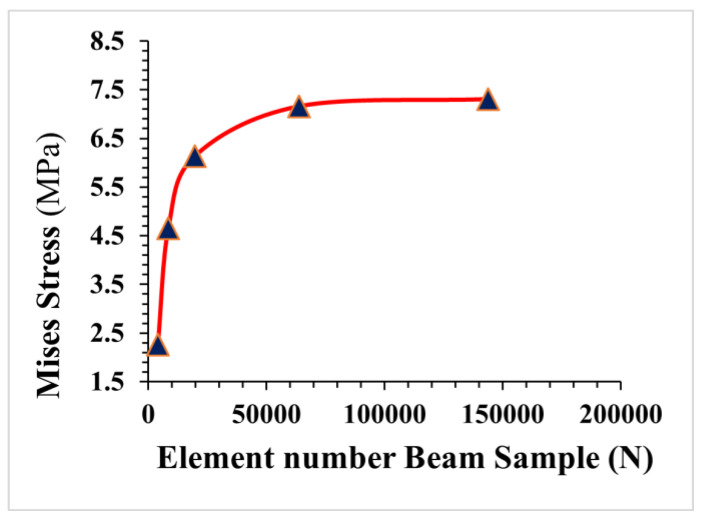
Mesh convergence analysis of FRC beam.

**Figure 10 materials-16-03700-f010:**
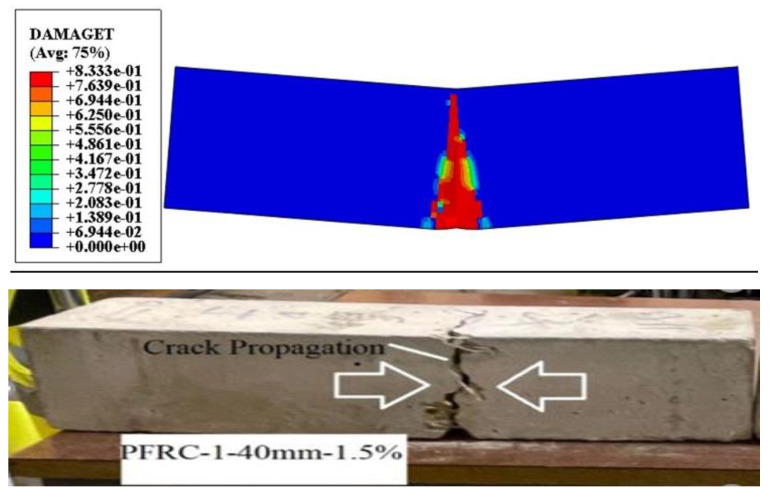
Comparison of the numerical and experimental results of PFRC-1 (40 mm) for flexural strength.

**Figure 11 materials-16-03700-f011:**
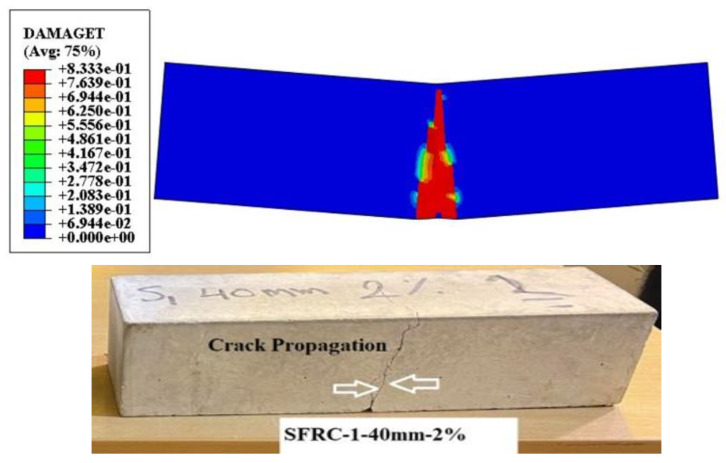
Comparison of the numerical and experimental results of SFRC-1 (40 mm) for flexural strength.

**Figure 12 materials-16-03700-f012:**
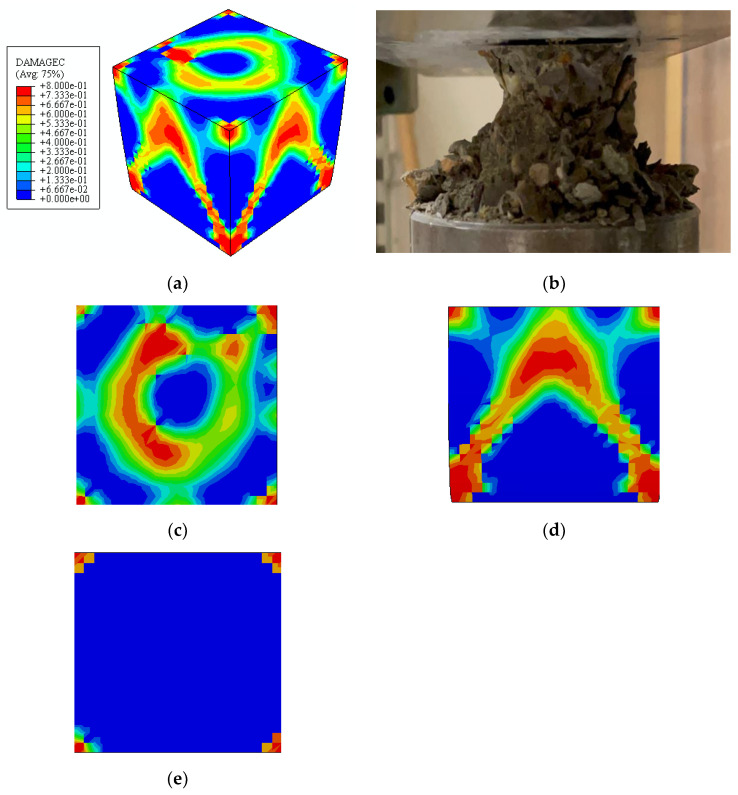
Comparison of the numerical and experimental results of the compressive strength behaviour of FRC. (**a**) damage modes of the experimental sample, (**b**) damage modes of the numerical sample, (**c**) damage modes at the (XZ TOP) (**d**) damage modes at the (YZ) (**e**) damage modes at the (XZ bottom).

**Figure 18 materials-16-03700-f018:**
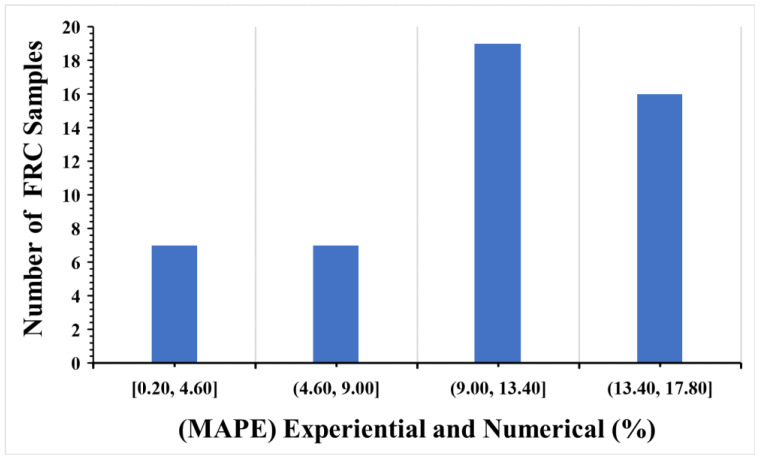
Mean absolute percentage error (MAPE) of flexural strength.

**Table 1 materials-16-03700-t001:** Properties of the fibres used in the study.

Type of Fibre	Shape	Diameter (mm)	Length of Fibre (mm)	Tensile Strength of Fibre (N/mm^2^)	Supplier
SFRC-1	Indented	1	(20, 30, 40)	1150	Sika
SFRC-2	dumbbell	1.45	(20, 30, 40)	690	Sika
PFRC-1	Macro/Monofilament	0.95	(20, 30, 40)	465	Sika
PFRC-2	Crimped	0.92	(20, 30, 40)	552	Sika

**Table 2 materials-16-03700-t002:** Mixing percentages of the control concrete.

Quantity	Cement (kg)	Water (kg)	Fine Aggregate (kg)	Coarse Aggregate (kg)
Per m^3^	427	213	679	1061
Trial mix 0.017 m^3^	7.26	3.62	11.54	18.037
Super-plasticizer (0.5% of cement)	0.0363 kg for trial mix			

**Table 3 materials-16-03700-t003:** Parameters of Concrete damage plasticity properties.

Dilation Angle	Eccentricity	fb0/fc0	K	Viscosity Parameter
36°	0.1	1.16	0.67	0

**Table 4 materials-16-03700-t004:** Concrete compressive and tensile behaviour.

Young’s Modulus MPa	34984	Poisson’s Ratio	0.2
Compression Behaviour		Concrete Compression Damage	
Stress (MPa)	Inelastic Strain	Damage Parameter	Inelastic Strain
16	0.000622996	0	0.000622996
20.01	0.000754057	0	0.000754057
24.00	0.014082286	0	0.014082286
28.01	0.015187781	0	0.015187781
32.02	0.015835257	0	0.015835257
36.00	0.017157429	0	0.017157429
38.93	0.020001589	0	0.020001589
38.72	0.021137714	0.005506698	0.021137714
35.89	0.022408446	0.078080053	0.022408446
26.70	0.024503275	0.314350292	0.024503275
15.60	0.028024185	0.599235637	0.028024185
7.79	0.033283509	0.799991781	0.033283509

**Table 5 materials-16-03700-t005:** Concrete tensile behavior.

Tensile Behaviour		Concrete Tension Damage	
Stress (MPa)	Cracking Strain	Damage Parameter	Cracking Strain
4.3678205	0.00014	0	0.00014
2.9118803	0.00042	0.333333333	0.00042
1.6379327	0.0008225	0.625	0.0008225
0.7279701	0.00147	0.833333333	0.00147

**Table 6 materials-16-03700-t006:** Properties of plastic fibers.

Plastic Fibres	
Young’s Modulus MPa	800
Poisson’s Ratio	0.33
Stress MPa	Strain
5.26	0.000064
20.00	0.000254
60.01	0.00073
100.13	0.001218
160.14	0.001948
200.10	0.002434
240.05	0.00292
274	0.003332

**Table 7 materials-16-03700-t007:** Properties of plastic fibers.

Steel Fibres	
Young’s Modulus MPa	200,000
Poisson’s Ratio	0.3
Stress MPa	Strain
98	0
195	0.0214844
309	0.022461
407	0.0234376
505	0.0385742
602	0.0551758
716	0.0703126
798	0.083496

**Table 9 materials-16-03700-t009:** Shows the flexural strength of fibre-reinforced concrete.

Mixes of FRC	Elastic Modulus of Fibre Types (MPa)	Tensile Strength of Different Type Shape (MPa)	Fibres Length (mm)	Percentage of Fibre (%)	Flexural Strength Experimental	Flexural Strength Numerical	(MSE) Experimental and Numerical	(MAPE) Experimental and Numerical (%)
					(MPa)	(MPa)		
SFRC-1	200,000	800	20	0.5	5.36	6.00	0.80	10.67
SFRC-1	200,000	800	20	1	5.50	6.10	0.77	9.84
SFRC-1	200,000	800	20	1.5	5.70	6.25	0.74	8.80
SFRC-1	200,000	800	20	2	5.30	6.00	0.84	11.67
SFRC-1	200,000	800	30	0.5	5.00	5.90	0.95	15.25
SFRC-1	200,000	800	30	1	6.10	6.30	0.45	3.17
SFRC-1	200,000	800	30	1.5	6.30	6.40	0.32	1.56
SFRC-1	200,000	800	30	2	5.50	6.05	0.74	9.09
SFRC-1	200,000	800	40	0.5	6.04	6.40	0.60	5.63
SFRC-1	200,000	800	40	1	5.31	6.10	0.89	12.95
SFRC-1	200,000	800	40	1.5	5.50	6.40	0.95	14.06
SFRC-1	200,000	800	40	2	5.73	6.36	0.79	9.91
PFRC-1	7000	465	20	0.5	5.30	6.36	1.03	16.67
PFRC-1	7000	465	20	1	5.80	6.50	0.84	10.77
PFRC-1	7000	465	20	1.5	5.78	6.40	0.79	9.69
PFRC-1	7000	465	20	2	5.30	6.06	0.87	12.54
PFRC-1	7000	465	30	0.5	5.70	6.47	0.88	11.90
PFRC-1	7000	465	30	1	5.43	6.20	0.88	12.42
PFRC-1	7000	465	30	1.5	5.57	6.29	0.85	11.45
PFRC-1	7000	465	30	2	6.00	6.45	0.67	6.98
PFRC-1	7000	465	40	0.5	5.20	6.10	0.95	14.75
PFRC-1	7000	465	40	1	5.10	6.05	0.97	15.70
PFRC-1	7000	465	40	1.5	5.50	6.00	0.71	8.33
PFRC-1	7000	465	40	2	5.30	5.95	0.81	10.92
PFRC-2	7000	552	20	0.5	5.21	6.05	0.92	13.88
PFRC-2	7000	552	20	1	5.70	6.45	0.87	11.63
PFRC-2	7000	552	20	1.5	6.20	6.41	0.46	3.28
PFRC-2	7000	552	20	2	5.05	5.90	0.92	14.41
PFRC-2	7000	552	30	0.5	5.92	6.47	0.74	8.50
PFRC-2	7000	552	30	1	5.40	6.15	0.87	12.20
PFRC-2	7000	552	30	1.5	5.50	6.20	0.84	11.29
PFRC-2	7000	552	30	2	5.73	6.50	0.88	11.85
PFRC-2	7000	552	40	0.5	5.56	6.34	0.88	12.30
PFRC-2	7000	552	40	1	5.50	6.25	0.87	12.00
PFRC-2	7000	552	40	1.5	5.75	6.30	0.74	8.73
PFRC-2	7000	552	40	2	5.60	6.38	0.88	12.23
SFRC-2	200,000	1150	20	0.5	5.42	6.10	0.82	11.15
SFRC-2	200,000	1150	20	1	5.50	6.28	0.88	12.42
SFRC-2	200,000	1150	20	1.5	5.40	6.12	0.85	11.76
SFRC-2	200,000	1150	20	2	5.20	6.00	0.89	13.33
SFRC-2	200,000	1150	30	0.5	5.10	5.95	0.92	14.29
SFRC-2	200,000	1150	30	1	6.10	6.29	0.44	3.02
SFRC-2	200,000	1150	30	1.5	6.30	6.40	0.32	1.56
SFRC-2	200,000	1150	30	2	5.90	6.10	0.45	3.28
SFRC-2	200,000	1150	40	0.5	6.04	6.27	0.48	3.67
SFRC-2	200,000	1150	40	1	5.21	6.10	0.94	14.59
SFRC-2	200,000	1150	40	1.5	5.50	6.15	0.81	10.57
SFRC-2	200,000	1150	40	2	5.73	6.25	0.72	8.32
Average							0.8	10.31

## Data Availability

Not applicable.

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
