# Peer review of "Modelling Fibre-Reinforced Concrete for Predicting Optimal Mechanical Properties"

_materials, 2023, doi:10.3390/ma16103700_

Round 1

Reviewer 1 Report

This paper provides a Khan Khalel model that can predict the desirable compressive and flexural strengths for any given values of key fibre parameters. It is a meaningful topic for the design of fibre-reinforced cementitious composites. There are some problems to be discussed including the followings:

1. In the paper only four parameters affecting mechanical properties are considered, such as the type, shape, length, and percentage of fibers. Are there any other parameters? For example, curing time, vibration conditions, etc? 

2. How can we overcome the influence of statistical randomness when there is only one specimen under each condition in the paper, or can we ignore it?

3. Is there sufficient basis for using the first-order polynomial shown in Equation (1) to describe the influence of each parameter?

4. In Figure 7, are there three test results without fibers? The results of the three specimens cannot be completely consistent.

5. The description of the test process is not sufficient.

6. How do you model the interaction between fibers and concrete in numerical simulation, and how are the parameters in the constitutive relationship model determined?

Author Response

Author’s Response to the Review of Manuscript ID: materials-2362639

We would like to thank the reviewer for their careful evaluation of our work for their detailed suggestions. We provide detailed answers to the various suggestions and comments of the reviewer.

This paper provides a Khan Khalel model that can predict the desirable compressive and flexural strengths for any given values of key fibre parameters. It is a meaningful topic for the design of fibre-reinforced cementitious composites. There are some problems to be discussed including the followings:

  1. In the paper only four parameters affecting mechanical properties are considered, such as the type, shape, length, and percentage of fibers. Are there any other parameters? For example, curing time, vibration conditions, etc?

√ Response:

The authors agree with the reviewer that the parameters such as curing time and vibration conditions do influence the mechanical properties of the FRC. However, in this study, we have focused on the four fibre parameters while keeping curing time and vibration conditions same for all the tested specimens. Therefore, the observed results can be directly mapped as an influence of the reinforced fibre parameters. The authors have considered the reviewer suggestion for the possible future work.

  1. How can we overcome the influence of statistical randomness when there is only one specimen under each condition in the paper, or can we ignore it?

√ Response:

We did three specimens for each mix, and the standard deviation error of compressive and flexural strength has been explained and highlighted in sections 4.1.1,4.1.2 respectively and shown in figures 7 and 8.

  1. Is there sufficient basis for using the first-order polynomial shown in Equation (1) to describe the influence of each parameter?

√ Response:

The majority of previous study were used the first-order polynomial to predict the mechanical concrete properties and they found it has good accuracy for the concrete compressive prediction[1–4].

“the researcher found that the sensitivity analysis on two different sets of input variables was performed for all the three models including MLR . Results indicate that the accuracy of the concrete compressive strength prediction is highly dependent on the number of input variables”

[1]       F. Khademi, Kjiu. Behfarnia, Evaluation of concrete compressive strength using artificial neural network and multiple linear regression models, (2016).

[2]       F. Khademi, M. Akbari, S.M. Jamal, M. Nikoo, Multiple linear regression, artificial neural network, and fuzzy logic prediction of 28 days compressive strength of concrete, Front. Struct. Civ. Eng. 11 (2017) 90–99.

[3]       F. Khademi, M. Akbari, S.M. Jamal, Prediction of compressive strength of concrete by data-driven models, I-Manager’s J Civ Eng. 5 (2015) 16–23.

[4]       S.M. Pandhiani, Assessment of coefficient of discharge of gabion weir using soft computing techniques, Int. J. Hydrol. Sci. Technol. 15 (2023) 249.

  1. In Figure 7, are there three test results without fibers? The results of the three specimens cannot be completely consistent.

√ Response:

           The tests were performed by using three samples for each mix. The figures 7 and 8 explain the   error. In addition, the figure has been modified and this sample for the control concrete without fiber.

  1. The description of the test process is not sufficient.

√ Response:

The description of the tests has been improved and explain in section 2.3.1. and 2.3.2.

  1. How do you model the interaction between fibers and concrete in numerical simulation, and how are the parameters in the constitutive relationship model determined?

√ Response:

The model interaction already explains and highlighted in section 3.1.

Embedding the fibres in the concrete region is assumed to lead to a perfect bond between the concrete and the fibres. It is worth mentioning that slipping behaviours have the same bond idea for both beam and cube. Despite this, the perfect bonding assumption has been widely utilized in the literature for concrete-like structures [72,73]. Digimat-FE software was used for the composite materials to obtain the random distribution of fibres inside the concrete samples[83].

Reviewer 2 Report

The manuscript, entitled "Modeling fiber-reinforced concrete for predicting optimal mechanical properties," presents an experimental study conducted on the modeling of flexural strength and compressive strength behavior of concrete reinforced with different types and amounts of fibers. However, the paper is way too long as it includes multiple unnecessary or redundant data points, the introduction section is way too vague, and many other issues should be addressed. The paper needs major revisions before it is processed further. Some comments follow:

Abstract: The abstract is written qualitatively. There are no comments about the results and future directions obtained from this study. There are only some general comments about the accuracy of the method. The abstract should be suitable for separate publication.

Introduction

The introduction should be significantly improved. Please clearly highlight the pros and cons of previous results and justify the need for the current research. Please discuss the highlights individually and assure a clear correspondence between the affirmations from the manuscript and those from the cited papers (the citations introduced in bulk form: ividually and assure a clear correspondence between the affirmations from the manuscript and those from the cited papers (the citations introduced in bulk form: "[1–5]" "[9–14]," etc.). Remove the bulk form by discussing each study separately. Please provide one short sentence for each manuscript that was cited that shows its relevance to the current study.

Methodology

Please rename this section as Materials and Methods.

"Figure 1 presents images of the different types of fibers used in this study." Please provide a 2D section of each type of fiber as a label for the figure part. Currently, the figure's shape and length are not clear.

Figure 4: The quality of the figure is poor; please provide a higher resolution image. Also, please change the color of the fibers so they can be visible in the beam.

Results and discussion

The standard deviation error should be provided for each measurement. These are multiphase/multicomponent materials; only one value or sample can be used for comparative evaluation. Some of the differences can be within the range of the error measurement.

Please approximate all the values from the graph, considering the error of the equipment, and provide the deviation bar in each graph.

Also, for the control sample, only one bar should be provided in each graph, as there are no fibers in the composition (currently, the association is ambiguous and incorrect).

All figures obtained from the modeling software are unclear; please provide higher resolution images.

Figure 11: Figures a and b are the same. Figure 11 should be removed as it doesn't provide any data of interest to the readers.

Same comments for Figure 13. Or move them to supplemental data.

Table 6 shows the compressive strength of fiber-reinforced concrete. Please check your experimental results. Elastic modulus of 2000 or tensile strength of 800 MPa for concrete is very surprising.

Conclusions section: Please improve the conclusions and present them following the main recommendations by academia of giving the conclusions of the study by points with highlights.

Future directions and limitations: Please provide some future directions and limitations of the study. This section is very important for this study, since the authors consider the aim of the study to be finding the optimal combination of four key parameters (shape, type, length, and percentage of fibers), but only four types of fiber and a very limited number of dimensions and shapes have been used in the experimental part. Therefore, it cannot be considered that this study covers the entire range of mixtures and fibers used previously to reinforce concrete.

English grammer

Please check the entire manuscript for spelling and typing errors.

For example, please remove the dot "." after the title.

Please check the entire manuscript for spelling and typing errors.

For example, please remove the dot "." after the title.

Also, there are multiple ambiguous formulations, especially in the titles of the tables and figures.

Author Response

Author’s Response to the Review of Manuscript ID: materials-2362639

We would like to thank the reviewer for their careful evaluation of our work for their detailed suggestions. We provide detailed answers to the various suggestions and comments of the reviewer.

The manuscript, entitled "Modeling fiber-reinforced concrete for predicting optimal mechanical properties," presents an experimental study conducted on the modeling of flexural strength and compressive strength behavior of concrete reinforced with different types and amounts of fibers. However, the paper is way too long as it includes multiple unnecessary or redundant data points, the introduction section is way too vague, and many other issues should be addressed. The paper needs major revisions before it is processed further. Some comments follow:

  1. Abstract:The abstract is written qualitatively. There are no comments about the results and future directions obtained from this study. There are only some general comments about the accuracy of the method. The abstract should be suitable for separate publication.

√ Response:

This has been updated and highlighted, we explain the results and the future work.

  1. Introduction

The introduction should be significantly improved. Please clearly highlight the pros and cons of previous results and justify the need for the current research. Please discuss the highlights individually and assure a clear correspondence between the affirmations from the manuscript and those from the cited papers (the citations introduced in bulk form: ividually and assure a clear correspondence between the affirmations from the manuscript and those from the cited papers (the citations introduced in bulk form: "[1–5]" "[9–14]," etc.). Remove the bulk form by discussing each study separately. Please provide one short sentence for each manuscript that was cited that shows its relevance to the current study.

√ Response:

  The bulk reference has been removed and we have added a short explanation for each reference.

  1. Methodology

√ Response:

  1. Please rename this section as Materials and Methods.

            The section has been renamed.

  1. "Figure 1 presents images of the different types of fibers used in this study." Please provide a 2D section of each type of fiber as a label for the figure part. Currently, the figure's shape and length are not clear.

  1. Figure 4: The quality of the figure is poor; please provide a higher resolution image. Also, please change the colour of the fibres so they can be visible in the beam.

Results and discussion

√ Response:

              The figures have been changed to a higher resolution.

  1. The standard deviation error should be provided for each measurement. These are multiphase/multicomponent materials; only one value or sample can be used for comparative evaluation. Some of the differences can be within the range of the error measurement.

Please approximate all the values from the graph, considering the error of the equipment, and provide the deviation bar in each graph.

√ Response:

  1. The standard deviation error of compressive and flexural strength has been explained and highlighted in sections 4.1.1,4.1.2 respectively and shown in figures 7 and 8.

Also, for the control sample, only one bar should be provided in each graph, as there are no fibers in the composition (currently, the association is ambiguous and incorrect).

√ Response:

The control sample has been added in the figure 7 and 8 for the mechanical test.

All figures obtained from the modeling software are unclear; please provide higher resolution images.

Figure 11: Figures a and b are the same. Figure 11 should be removed as it doesn't provide any data of interest to the readers.

√ Response:

The figure 11 has been removed.

Same comments for Figure 13. Or move them to supplemental data.

√ Response:

The figure 13 has been removed.

Table 6 shows the compressive strength of fibre-reinforced concrete. Please check your experimental results. Elastic modulus of 20000 or tensile strength of 800 MPa for concrete is very surprising.

Conclusions section: Please improve the conclusions and present them following the main recommendations by academia of giving the conclusions of the study by points with highlights.

√ Response:

These elastic modulus in the table were for the fibre that used in this study is not for the concrete.

The conclusion has been improved and highlights the significant points.

Future directions and limitations: Please provide some future directions and limitations of the study. This section is very important for this study, since the authors consider the aim of the study to be finding the optimal combination of four key parameters (shape, type, length, and percentage of fibers), but only four types of fiber and a very limited number of dimensions and shapes have been used in the experimental part. Therefore, it cannot be considered that this study covers the entire range of mixtures and fibers used previously to reinforce concrete.

The future direction and limitation has been added and explain in conclusion section.

“The proposed model has used elastic modulus for material input selection and hence predominantly considers linear elastic behaviour in FRC during tests. However, a use of plasticity correction on this input can represent the complex nonlinear relationship between the various input variables and properties of FRC. Also, should be consider the different shapes and more dimensions of the fibre . This will be considered as a change to be implemented in future work.”

References

English grammar

√ Response:

Please check the entire manuscript for spelling and typing errors.

For example, please remove the dot "." after the title.

 Also, there are multiple ambiguous formulations, especially in the titles of the tables and figures

√ Response:

The spelling has been checked and the dot “.” After the title has been removed. Also, the ambiguous formulations have been checked and highlighted.

Reviewer 3 Report

This paper proposed a novel model in which some key fibre parameters was taken as inputs to predict the compressive strength and the flexural strength of reinforced concrete. The results are interesting and attractive.It can be accepted after a major revision.

(1)       Line75, “Granulated blast furnace slag was used as a fibre.” This sentence is very difficult to understand.

(2)       Table 4 and Table 5 should be redesigned.

(3)       The vertical scale of the four figures in Fig.8 should be maintained consistent.

(4)       The data in Fig.15 and Fig.18 show bad dispersibility. The linear relationship between flexural strength results for the experimental and numerical models is not good. Why the correlation coefficients both are as high as 0.99 and more?

(5)       What are the physical meanings of the equations (16) and (17)? Each parameter should be assigned a symbol and unit.

(6)       In Fig.18, the word “Numrical” should be changed to Numerical.

(7)       The title of Table 7 to Table 9 is not suitable.

(8)       In the conclusions, the accuracy of the prediction model should be improved since the maximum error of 15% is a little high.

(9)       Some related studies can improve the work especially the introduction part. Such as: Influences of MgO and PVA fiber on the abrasion and cracking resistance, pore structure and fractal features of hydraulic concrete; Comparison of fly ash, PVA fiber, MgO and shrinkage-reducing admixture on the frost resistance of face slab concrete via pore structural and fractal analysis

The Quality of English is fine.

Author Response

Author’s Response to the Review of Manuscript ID: materials-2362639

We would like to thank the reviewer for their careful evaluation of our work for their detailed suggestions. We provide detailed answers to the various suggestions and comments of the reviewer.

This paper proposed a novel model in which some key fiber parameters was taken as inputs to predict the compressive strength and the flexural strength of reinforced concrete. The results are interesting and attractive. It can be accepted after a major revision.

(1)       Line75, “Granulated blast furnace slag was used as a fiber.” This sentence is very difficult to understand.

√ Response:

I agree with you the sentence has been deleted.

(2)       Table 4 and Table 5 should be redesigned.

√ Response:

The tables 4 and 5 has been redesigned.

(3)       The vertical scale of the four figures in Fig.8 should be maintained consistent.

√ Response:

The vertical scale has been checked.

(4)       The data in Fig.15 and Fig.18 show bad dispersibility. The linear relationship between flexural strength results for the experimental and numerical models is not good. Why the correlation coefficients both are as high as 0.99 and more.

√ Response:

The correct R2 value are now inserted in the figure. The selected linear polynomial seems poor fit, but it is fine to predict the values of compressive and flexural strengths as the standard deviation is within a range of (0,03 to 2.6) and (0.04 to 0.8) MPa.

(5)       What are the physical meanings of the equations (16) and (17)? Each parameter should be assigned a symbol and unit.

√ Response:

The physical meanings have been explained and highlighted. Also the units has been explained.

(6)       In Fig.18, the word “Numrical” should be changed to Numerical.

√ Response:

The mistake has been in Fig 18 corrected.

(7)       The title of Table 7 to Table 9 is not suitable.

√ Response

The title of the tables from 7 to 9 has been rewritten.

(8)       In the conclusions, the accuracy of the prediction model should be improved since the maximum error of 15% is a little high.

√ Response:

As I mention in the manuscript in section 4.3.” According to the MAPE evaluated tool results, only four of the numerical values in the CDP model showed an error (16.67%,15.70,14.75%14.59%) between all experimental concrete flexural strength output values (48 experimental findings). Also, the majority of errors between 1-12% which is acceptable in concrete modeling”.

“The overall variation in the test results is ranging from 3.84%-12%. It is quite acceptable specially in the presence of extensive diversity in test specimens, materials, loading configurations, experimental methodologies, and test arrangements [100,101].”

(9)       Some related studies can improve the work especially the introduction part. Such as: Influences of MgO and PVA fiber on the abrasion and cracking resistance, pore structure and fractal features of hydraulic concrete; Comparison of fly ash, PVA fiber, MgO and shrinkage-reducing admixture on the frost resistance of face slab concrete via pore structural and fractal analysis

√ Response:

The related study you were recommended has been added and highlighted.

Round 2

Reviewer 2 Report

Dear authors, 

You have done a good job revising the paper. Further, please provide higher-resolution images for Figure 4, Figure 5, and Figures 7 to 10. These figures are of poor quality, and they cannot be published in their current form.

Best regards.

Author Response

Author’s Response to the Review of Manuscript ID: materials-2362639

We would like to thank the reviewer for their careful evaluation of our work for their detailed suggestions. We provide detailed answers to the various suggestions and comments of the reviewer.

You have done a good job revising the paper. Further, please provide higher-resolution images for Figure 4, Figure 5, and Figures 7 to 10. These figures are of poor quality, and they cannot be published in their current form.

√ Response:

All the figures have been modified to a good resolution.